# Approaching coupled cluster accuracy with a general-purpose neural network potential through transfer learning

Justin S. Smith[1,2,3,7], Benjamin T. Nebgen [2,4,7], Roman Zubatyuk[2,5,7], Nicholas Lubbers[2,3], Christian Devereux[1], Kipton Barros[2], Sergei Tretiak [2,4], Olexandr Isayev [6] & Adrian E. Roitberg[1]

Computational modeling of chemical and biological systems at atomic resolution is a crucial tool in the chemist's toolset. The use of computer simulations requires a balance between cost and accuracy: quantum-mechanical methods provide high accuracy but are computationally expensive and scale poorly to large systems, while classical force fields are cheap and scalable, but lack transferability to new systems. Machine learning can be used to achieve the best of both approaches. Here we train a general-purpose neural network potential (ANI-1ccx) that approaches CCSD(T)/CBS accuracy on benchmarks for reaction thermochemistry, isomerization, and drug-like molecular torsions. This is achieved by training a network to DFT data then using transfer learning techniques to retrain on a dataset of gold standard QM calculations (CCSD(T)/CBS) that optimally spans chemical space. The resulting potential is broadly applicable to materials science, biology, and chemistry, and billions of times faster than CCSD(T)/CBS calculations.

[1] Department of Chemistry, University of Florida, Gainesville, FL 32611, USA. [2] Theoretical Division, Los Alamos National Laboratory, Los Alamos, NM 87545, USA. [3] Center for Nonlinear Studies, Los Alamos National Laboratory, Los Alamos, NM 87545, USA. [4] Center for Integrated Nanotechnologies, Los Alamos National Laboratory, Los Alamos, NM 87545, USA. [5] Department of Chemistry, Physics, and Atmospheric Science, Jackson State University, Jackson, MS 39217, USA. [6] UNC Eshelman School of Pharmacy, University of North Carolina at Chapel Hill, Chapel Hill, NC 27599, USA. [7] These authors contributed equally: Justin S. Smith, Benjamin T. Nebgen, Roman Zubatyuk. Correspondence and requests for materials should be addressed to S.T. (email: serg@lanl.gov) or to O.I. (email: olexandr@olexandrisayev.com) or to A.E.R. (email: roitberg@ufl.edu)

The central questions in modern chemistry relate to the identification and synthesis of molecules for useful applications. Historically, discoveries have often been serendipitous, driven by a combination of intuition and experimental trial and error[1,2]. In the modern age, the computer revolution has brought about powerful computational methods based on quantum mechanics (QM) to create a new paradigm for chemistry research[3,4]. At great computational expense, these methods can provide accurate chemical properties (e.g., energies, forces, structures, reactivity, etc.) for a wide range of molecular systems. Coupled-cluster theory systematically approaches the exact solution to the Schrödinger equation, and is considered a gold standard for many quantum chemistry applications[5–7]. When CCSD (T) (coupled cluster considering single, double, and perturbative triple excitations) calculations are combined with an extrapolation to the complete basis set limit (CBS)[8,9], even the hardest to predict non-covalent and intermolecular interactions can be computed quantitatively[10]. However, coupled-cluster theory at the level of CCSD(T)/CBS is computationally expensive, and often impractical for systems with more than a dozen atoms.

Since the computational cost of highly accurate QM methods can be impractical, researchers often seek to trade accuracy for speed. Density functional theory (DFT)[11–13], perhaps the most popular QM method, is much faster than coupled-cluster theory. In practice, however, DFT requires empirical selection of a density functional, and so DFT-computed properties are not as reliable and objective as coupled-cluster techniques at guiding experimental science. Even stronger approximations can be made to achieve better efficiency. For example, classical force fields are commonly employed to enable large scale dynamical simulation such as protein folding[14], ligand-protein docking[15], or the dynamics of dislocations in materials[16]. These models are often fragile; a force field fit to one system may not accurately model other systems[17]. An outstanding challenge is to simultaneously capture a great diversity of chemical processes with a single linear-scaling model potential.

Machine learning (ML) methods have seen much success in the last decade due to increased availability of data and improved algorithms[18–20]. Applications of ML are becoming increasingly common in experimental and computational chemistry. Recent chemistry related work reports on ML models for chemical reactions[21,22], potential energy surfaces[23–27], forces[28–30], atomization energies[31–33], atomic partial charges[32,34–36], molecular dipoles[26,37,38], materials discovery[39–41], and protein-ligand complex scoring[42]. Many of these studies represent important and continued progress toward ML models of quantum chemistry that are transferable (i.e., applicable to related, but new chemical processes) and extensible (i.e., accurate when applied to larger systems). These advances aim to revolutionize chemistry through applications to chemical and biological systems. Since molecular dynamics simulations underpin much of computational chemistry and biology, transferable, accurate, and fast prediction of molecular energies and forces is particularly important for the next generation of linear-scaling model potential energy surfaces.

Transferable and extensible ML potentials often require training on very large data sets. One such approach is the ANI class of methods. The ANI-1 potential aims to work broadly for molecules in organic chemistry[43]. A key component of this potential is the ANI-1 data set, which consists of DFT energies for 22M randomly selected molecular conformations from 57k distinct small molecules[44]. This vast amount of data would be impractical to generate at a level of theory more accurate than DFT[45]. However, advances in machine learning methodologies are greatly reducing the required data set sizes. The ANI-1x data set, constructed using active learning, contains DFT data for 5M conformations of molecules with an average size of 15 atoms[25].

Active learning iteratively adds new QM calculations to the data set for specific cases where the current ML model cannot make a good prediction. Despite the much smaller size of the ANI-1x data set, potentials trained on it vastly outperform those trained on the ANI-1 data set, especially on transferability and extensibility benchmarks. Even with the success of the ANI-1x potential, its true accuracy is still reliant upon the accuracy of the underlying DFT data.

A remaining challenge is to develop ML-based potentials that reach coupled-cluster-level accuracy while retaining transferability and extensibility over a broad chemical space. The difficulty is that data sets with CCSD(T)-level accuracy are very expensive to construct and therefore tend to be limited in chemical diversity. Previous studies have trained on high-quality QM data for small molecules at equilibrium conformations[46,47] and for non-equilibrium conformations of a single molecule[48]. A limitation is that ML models trained on data sets which lack chemical diversity are not expected to be transferable or extensible to new systems. The present work uses transfer learning[49,50] to train an ML potential that is accurate, transferable, extensible, and therefore, broadly applicable. In transfer learning, one begins with a model trained on data from one task and then retrains the model on data from a different, but related task, often yielding high-accuracy predictions[51–53] even when data are sparsely available. In our application, we begin by training a neural network on a large quantity of lower-accuracy DFT data (the ANI-1x data set with 5 M non-equilibrium molecular conformations[25]), and then we retrain to a much smaller data set (about 500k intelligently selected conformations from ANI-1x) at the CCSD (T)/CBS level of accuracy. Such a high-quality and diverse data set is a first of its kind for training machine learning-based potentials. The resulting general-purpose potential, ANI-1ccx, and data set exceeds the accuracy of DFT in benchmarks for isomerization energies, reaction energies, molecular torsion profiles, and energies and forces at non-equilibrium geometries, while being roughly nine orders of magnitude faster than DFT. The ANI-1ccx potential is available on GitHub (https://github.com/isayev/ASE_ANI) as a user-friendly Python interface integrated with the Atomic Simulation Environment[54] package (ASE; https://wiki.fysik.dtu.dk/ase/).

## Results

**Relative conformer energy**. We compare the errors of ANI-1ccx (trained with transfer learning), ANI-1x (trained on DFT data only), and direct DFT calculations (ωB97X/6-31g*). We also compare to a model, ANI-1ccx-R, that was trained only with the CCSD(T)*/CBS data, i.e., without transfer learning from the DFT data. CCSD(T)*/CBS is a highly accurate extrapolation to high level QM. For details see the methods section. To test transferability and extensibility, we employ four benchmarks to appraise the accuracy of molecular energies and forces, reaction thermochemistry, and the computation of torsional profiles on systems consisting of CHNO. The GDB-10to13 benchmark[25] is designed to evaluate relative energies, atomization energies, and force calculations on a random sample of 2996 molecules containing 10–13 C, N, or O atoms (with H added to saturate the molecules). The GDB-10to13 molecules are randomly perturbed along their normal modes to produce between 12 and 24 non-equilibrium conformations per molecule. HC7/11[55] is a benchmark designed to gauge the accuracy of hydrocarbon reaction and isomerization energies. The ISOL6 benchmark[56] (a subset of the ISOL24/11 benchmark) measures isomerization energies for organic molecules. Finally, we test on the Genentech torsion benchmark[57], which contains 62 diverse organic molecule torsion profiles (45 containing only CHNO).

**Table 1 Accuracy in predicting conformer energy differences on the GDB-10to13 benchmark**

|  | ANI-1ccx | ANI-1ccx-R | ANI-1x | ωB97X |
|---|---|---|---|---|
| MAD[a] | 1.46 | 1.81 | 1.97 | 1.42 |
| RMSD[a] | 2.07 | 2.54 | 2.79 | 2.04 |

[a]Units are in kcal mol$^{-1}$

Table 1 provides mean absolute deviations (MAD) and root mean squared deviations (RMSD) for the ANI potentials and ωB97X/6-31g*, on the GDB-10to13 benchmark from the COMP6[25] benchmark suite. Reference values are recomputed at the CCSD(T)*/CBS level of theory. Table 1 only considers conformations within 100 kcal mol$^{-1}$ of the energy minima for each molecule. The conformational energy $\Delta E$ is the energy difference between all conformers for a given molecule in the benchmark[25]. Methods compared are the ANI-1ccx transfer learning potential, ANI-1ccx-R trained only on coupled-cluster data, ANI-1x trained only on DFT data, and the DFT reference (ωB97X). Our analysis concludes that training a model only to the smaller CCSD(T)*/CBS data set (ANI-1ccx-R) results in a 23% degradation in RMSD compared with the transfer learning model (ANI-1ccx). The DFT trained ANI-1x model has a 36% increase in RMSD over ANI-1ccx. ANI-1ccx performs as well as the original reference (ωB97X/6-31G*) in the 100 kcal mol$^{-1}$ energy range on the GDB-10to13 CCSD(T)*/CBS benchmark. Recall that each ANI model is an ensemble average over eight neural networks. Without an ensemble of networks, the MAD and RMSD of ANI models degrades by about 25%[25]. Supplementary Table 5 provides errors for all methods within the full energy range of the GDB-10to13 benchmark. Notably, ANI-1ccx outperforms DFT with an RMSD of 3.2 kcal mol$^{-1}$ vs. 5.0 kcal mol$^{-1}$ for DFT, which means the ANI-1ccx model generalizes better to high energy conformations than ωB97X/6-31G*. Supplementary Fig. 3 shows correlation plots for the ANI models vs. CCSD(T)*/CBS.

**Atomization energy**. Figure 1 displays a comparison of atomization energy deviation from reference CCSD(T)*/CBS for DFT (blue) and ANI-1ccx (orange) for all conformations in GDB-10to13 within 100 kcal mol$^{-1}$ of the conformational minima. Compared with the DFT functional, the ANI-1ccx potential provides a more accurate prediction of the CCSD(T)*/CBS atomization energy. The distribution for ANI-1ccx has a standard deviation of 2.3 kcal mol$^{-1}$, while the DFT distribution is much wider, with a standard deviation of 6.3 kcal mol$^{-1}$. The MAD/RMSD for DFT vs. reference CCSD(T)*/CBS is 15.9/17.1 kcal mol$^{-1}$, while for ANI-1ccx it is 1.9/2.5 kcal mol$^{-1}$. Supplementary Fig. 4 shows an attempt to correct the systematic shift of the DFT model to the reference CCSD(T)*/CBS atomization energies via a linear fitting of the atomic elements in each system. Even after this non-trivial correction, ANI-1ccx is still more accurate than DFT vs. the more accurate coupled-cluster atomization energies. The corrected DFT has a distribution with a standard deviation of 5.5 kcal mol$^{-1}$ with MAD/RMSD of 4.9/5.9 kcal mol$^{-1}$.

**Forces**. Accurate forces are important for MD simulations and geometry optimization. Therefore, we explicitly assess force accuracy as well. It is impractical to obtain forces with the CCSD(T)*/CBS extrapolation due to extreme computational expense with existing packages. However, MP2/cc-pVTZ (dubbed here as MP2/TZ) provides a high-quality alternative. Table 2 compares MP2/TZ force calculations on the GDB-10to13 benchmark to MP2/cc-pVDZ (MP2/DZ), ωB97X/6-31G*, ANI-1x, and ANI-

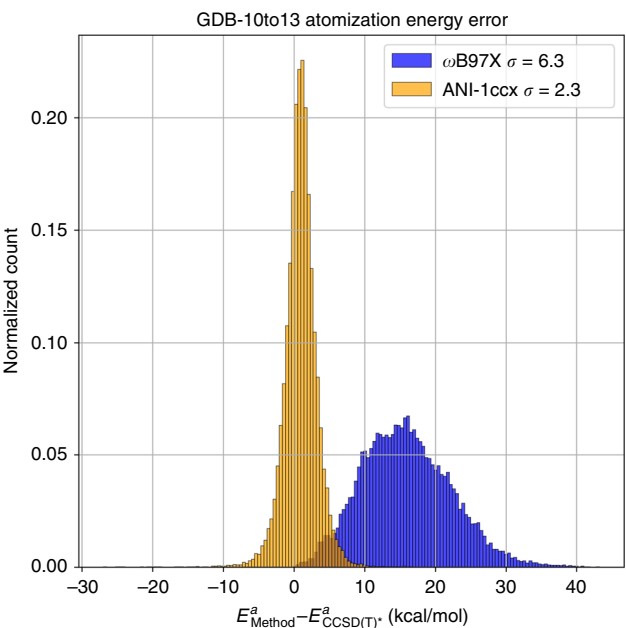

**Fig. 1** Accuracy in predicting atomization energies. Error of the ANI-1ccx predicted atomization energy $E^a$ on the GDB-10to13 benchmark relative to CCSD(T)*/CBS and compared against ωB97X

**Table 2 Accuracy for calculating atomic forces on the GDB-10to13 benchmark**

|  | ANI-1ccx | ANI-1x | ωB97X | MP2/DZ |
|---|---|---|---|---|
| MP2/TZ | 3.4/5.3[a] | 4.7/7.1[a] | 3.7/5.9[a] | 4.6/5.9[a] |

[a]MAE/RMSE in kcal mol$^{-1}$ Å$^{-1}$

1ccx models. ANI-1ccx provides the best prediction of MP2/TZ forces compared with all other methods. Notably, ANI-1ccx forces deviate less from the MP2/TZ target forces than the original ANI-1x DFT trained potential, providing evidence that the transfer learning process not only corrects energies but forces as well. Supplementary Fig. 5 also shows a comparison between ANI-1ccx and experimental results for C-C center of mass radial distribution functions for cyclohexane.

**Reaction and isomerization energy**. The HC7/11 and ISOL6 benchmarks address the calculation of reaction and isomerization energies and are depicted in Fig. 2. For each reaction, reference energies and calculated energies are provided in Supplementary Tables 7 and 8. Figure 2 shows the differences between the computed and the reference energies, for the reaction and isomerization energies individually for ωB97X/6-31g*, ANI-1x, ANI-1ccx, and our CCSD(T)*/CBS. HC7/11 used target MP2/6-311+G(2df,2p) and ISOL6 used target CCSD(T)-F12a/aug-cc-pVDZ calculations. The latter is the most accurate simulation method currently available. As was done in the original benchmarks, single point energy calculations using all ANI models, ωB97X, and CCSD(T)*/CBS were performed on the original benchmark structures. These energies were used to calculate the reaction energies. For the HC7/11 benchmark, the medium-sized basis DFT reference ωB97X/6-31g* is not sufficient for describing the chemistry represented in these complex hydrocarbon reactions. Likewise, ANI-1x, trained to data from

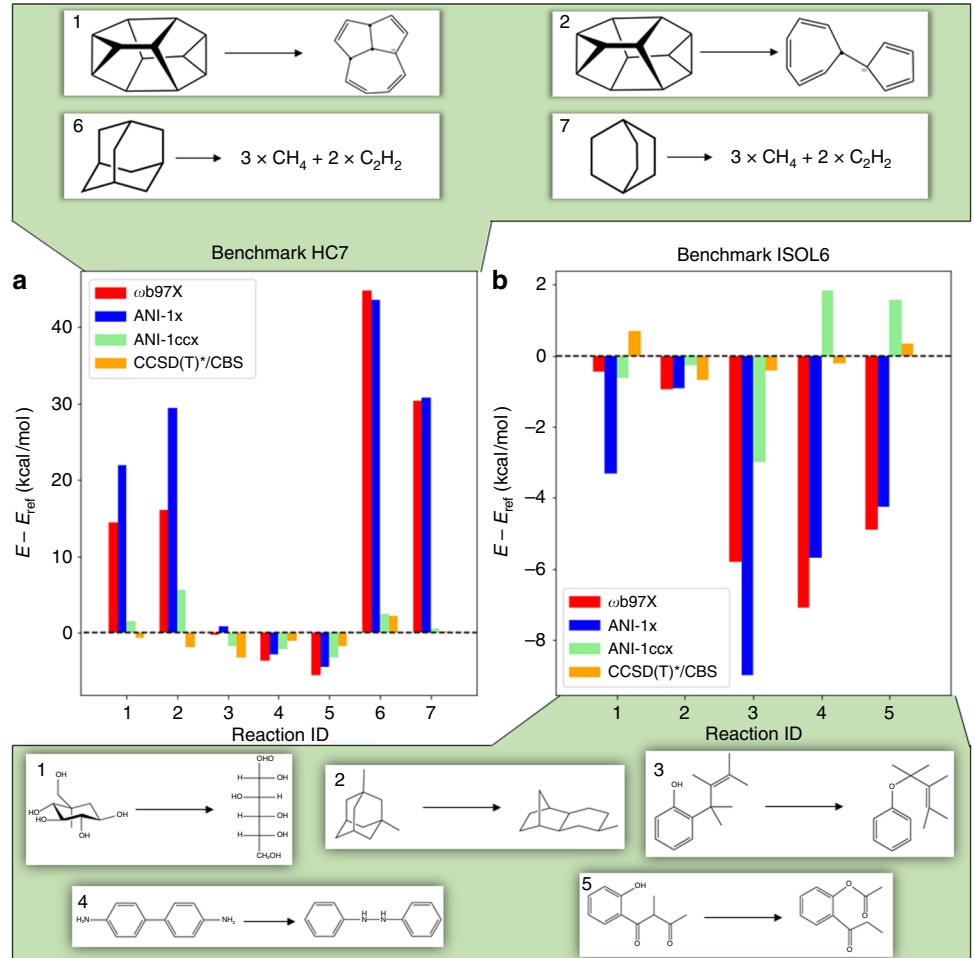

**Fig. 2** Accuracy in predicting reaction and isomerization energy. ANI-1ccx reaction and isomerization energy difference prediction on the **a** HC7/11 and **b** ISOL6 benchmarks, relative to CCSD(T)/CBS. Methods compared are the ANI-1ccx transfer learning potential, ANI-1x trained only on DFT data, the DFT reference (ωB97X), and our coupled-cluster extrapolation scheme CCSD(T)*/CBS. The top panel provides the HC7 reactions numbered 1, 2, 6, and 7 and bottom panel shows the ISOL6 reactions numbered 1–5

this functional, closely mirrors the behavior of DFT. Similarly, the transfer learning-based ANI-1ccx model tends to mirror its CCSD(T)*/CBS reference calculations and substantially outperforms DFT compared with the target reaction energies. Overall MAD/RMSD on the HC7/11 benchmark for DFT, ANI-1x, ANI-1ccx, and CCSD(T)*/CBS are 16.4/22.2, 19.1/24.6, 2.5/2.9, and 1.6/1.8 kcal mol$^{-1}$, respectively. These results are shown in Supplementary Table 6.

Figure 2b displays a similar comparison for the five medium-sized organic C, H, N, O containing molecules of the ISOL6[56] isomerization energy benchmark. A similar trend is seen in this case as with the HC7/11 benchmark, where ANI-1x deviations tend to correlate with the large deviations of its reference DFT. The prediction error for the transfer learning-based model is greatly reduced compared with the DFT trained model and DFT itself. For the ISOL6 reactions shown in Fig. 2b, overall MAD/RMSD for DFT, ANI-1x, ANI-1ccx, and CCSD(T)*/CBS are 3.8/4.7, 4.6/5.3, 1.5/1.8, and 0.5/0.5 kcal mol$^{-1}$, respectively.

**Molecular torsions**. Molecular torsions play an import role in computational drug discovery (e.g., in screening ligands for favorable protein binding) and in modeling the assembly of soft materials. Therefore, we compare the new ANI-1ccx transfer learning-based potential against various QM and molecular

mechanics (MM) based methods from the molecular torsion benchmark of Sellers et al.[57]. This benchmark provides a measure of accuracy for a model at reproducing potential energy profiles from a diverse set of molecular torsions of small organic molecules containing the atoms C, H, N, and O. These torsions are representative torsions typically found in small drug-like molecules.

Figure 3 provides a comparison of results for three highly accurate but computationally expensive QM methods, four moderately computationally expensive QM methods, and two commonly used small-molecule force fields. These data were obtained from Sellers et al.[57]. We also add the ANI potentials (ANI-1ccx, ANI-1ccx-R, and ANI-1x) used in this work, as well as CCSD(T)*/CBS reference energy calculations. Other semi-empirical QM and MM methods studied in Sellers et al. are left out of this comparison since each one performed worse than OPLS2005 on the benchmark. Each torsion in the benchmark is generated through a restrained optimization, where the torsional degree of freedom is fixed every 10°, and the remaining degrees of freedom are relaxed through an optimization process. The red boxes in Fig. 3 represent QM methods (first three from the left) that are so computationally intense the MP2 restrained optimized structures were used and single point calculations were performed for that method. The green QM methods were all optimized using their own forces. In an ideal setting, all methods would provide

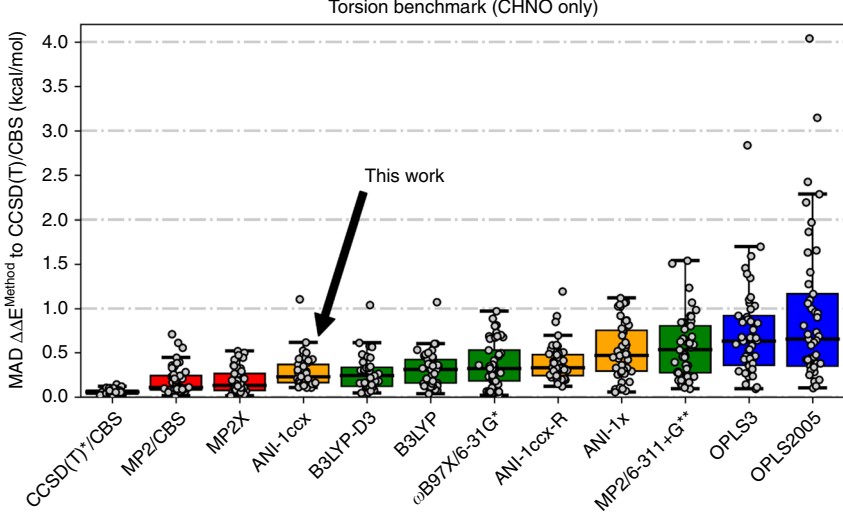

**Fig. 3** Accuracy in predicting torsional energies relevant to drug discovery. Methods compared are QM (red and green), molecular mechanics (blue), and ANI (orange) performance on 45 torsion profiles containing C, H, N, and O atomic elements. The gray dots represent the MAD of a given torsion scan vs. gold standard CCSD(T)/CBS. The box extends from the upper to lower quartile and the black horizontal line in the box is the median. The upper "whisker" extends to the last datum less than the third quartile plus 1.5 times the interquartile range while the lower "whisker" extends to the first datum greater than the first quartile minus 1.5 times the interquartile range

their own structures for the final energy calculation, though, such an exercise would be prohibitively computationally expensive for the more rigorous QM methods. The ANI and MM models also carried out restrained optimizations using their own forces. The MP2 structures are not used here because the usefulness of such efficient methods can only be gauged without the assistance of less efficient QM methods. The ANI-1x potential, trained to the ANI-1x DFT data set plus active learning-based dihedral corrections, obtains a median MAD of 0.47 kcal mol$^{-1}$ on the benchmark. The ANI-1x potential performs similarly to MP2/6-311+G** and to the ANI-1ccx-R potential. The DFT trained ANI-1x also outperforms OPLS3, one of the most accurate and widely used small-molecule force fields available. Further, the transfer learning-based ANI-1ccx potential achieves a median MAD of 0.23 kcal mol$^{-1}$, a 51% reduction in error over ANI-1x vs. the CCSD(T)/CBS target. ANI-1ccx exceeds the performance of all DFT (B3LYP-D3/6-311+G**, B3LYP/6-311+G**, and ωB97X/6-31g*) methods utilized in this study, approaching the accuracy of higher-level, and costlier, ab initio QM methods (MP2/CBS and MP2.X/CBS). The ANI-1ccx potential achieves these prediction accuracies without an increase in computational cost over the original ANI-1x potential. Results for ANI-1x and ANI-1ccx before and after active learning for dihedral reparameterization can be found in Supplementary Fig. 1. The dihedral scans used to compute ANI-1ccx's error can be found in Supplementary Fig. 6. Each ANI-1ccx restrained optimization (averaged over the 36 angles for each of the 45 torsions) took ~0.58 s on a single NVIDIA V100 GPU. A similar timing comparison was reported in Sellers et al.[57] for the QM and MM methods. Compared with this literature result, the ANI model on a single GPU is (on average) as fast as OPLS3 on a CPU and 6200 times faster than B3LYP-D3 on a CPU. While a GPU to CPU comparison with the use of different optimization methods is not exactly a fair comparison, it does provide a sense of the computational affordability of the ANI potential. Moreover, the ANI potential scales more easily than QM, exhibiting linear scaling (compared with O(N$^3$) for the most efficient QM) on large systems and a smaller pre-factor and better memory scaling.

## Discussion

Great progress has been made in creating faster and more accurate QM methods, but even in modern computer architectures the cost involved in the improved accuracy becomes prohibitive very quickly. With the advent of machine learning, we can and must make the leap to modern statistical and data-driven approaches, which have the potential to drive rapid progress in drug and materials design as well as applications to natural systems such as proteins. The ANI-1ccx potential (available at https://github.com/isayev/ASE_ANI) presented in this work is an attractive alternative to density functional theory approaches and standard force fields for conformational searches, molecular dynamics, and the calculation of reaction energies. The availability of high-quality QM reference data, produced with a new extrapolation scheme to CCSD(T)/CBS, allowed us to use transfer learning techniques to build a chemically accurate universal ANI potential. Accuracy benchmarks show that the transfer learning-based ANI-1ccx outperforms DFT on test cases where DFT fails to accurately describe reaction thermochemistry and on small-molecule torsion benchmarks. After extensive benchmarking, we conclude that ANI-1ccx captures a broad range of organic chemistry, with accuracy comparable to QM calculations at the coupled-cluster level of theory. Comparisons between transfer learning and naive training to only the small data set of high-quality QM calculations show that transfer learning is a superior approach. As such this work offers a computationally efficient and accurate ML-based molecular potential for general use across a broad range of chemical systems.

Future work will aim to validate and retrain (if necessary) the ANI-1ccx potential for applications in condensed phase simulation. For smaller molecular systems, the ANI-1ccx potential is an accurate and efficient alternative to expensive QM methods and might find indirect ways to become applicable in such condensed phase simulation, e.g., using ANI-1ccx to parametrize force fields for condensed phase simulation. As with any model, ani-1ccx has limitations. Some of them can be overcome by adding more data and through active learning methods and retraining. This category includes new and different chemical environments, inter-molecular interactions, ions, new atomic elements and reactions.

There is a set of limitations that would require the development of new theory and methods, for instance for recovering long-range interactions through the addition of coulomb interactions, to treat multiple electronic excited states or radicals.

## Methods

**An efficient and accurate CCSD(T)/CBS approximation.** Recalculating even 10% of the ANI-1x data set (i.e., 500k molecules) with conventional CCSD(T)/CBS would require enormous computational resources. Therefore, we developed an approximation scheme (herein referred to as CCSD(T)*/CBS) that allows highly accurate energy calculations in a high-throughput fashion.

Our CCSD(T)*/CBS method is a computationally efficient approximation of CCSD(T)/CBS energies that takes advantage of the linear-scaling domain-localized DPLNO-CCSD(T) method developed by Neese et al.[58] which is implemented in the ORCA software package[59]. It provides an affordable alternative capable of achieving near CCSD(T) accuracy at a fraction of the computational cost. The DLPNO approximation relies on the MP2 method to estimate energy contributions from interacting electron pairs and effectively reduce the active orbital space. Table 3 provides accuracy and timing benchmarks, clearly showing our CCSD(T)*/CBS approximation provides accurate energies vs. the CCSD(T)-F12 level of theory[60] in a computationally efficient way. S66 and W4-11 are standard benchmarks for interaction and atomization energies of small molecules[61,62]. See Supplementary Table 1 for a more detailed comparison. Details of our CCSD(T)*/CBS scheme plus additional benchmarks are given in supplemental information Section S1.1.

**Using active learning for CCSD(T)*/CBS data set curation.** The existing ANI-1x active learning generated data set[25] is used to train an initial DFT (to the $\omega$B97X/6-31G* model chemistry[63]) potential, likewise dubbed ANI-1x. The ANI-1x data set consists of 5M conformations from 64k small molecules and complexes of molecules containing only CHNO atoms. All model and training procedures are detailed in the ANI-1 work[43]. Section S1.2 provides details of the architecture, selection of hyperparameters, and held out test set errors. To reduce variance and increase accuracy, all ANI results presented in this work are the ensemble

prediction of eight ANI neural networks, i.e., the ANI-1x potential used in this work is an ensemble of eight ANI-1x neural networks trained to different splits of the ANI-1x data set and the ANI-1ccx network is built by transfer learning from the eight ANI-1x networks[25]. The disagreement between predictions of ensemble members can be used as a proxy to the prediction error, enabling rapid identification of molecular conformations where the current ANI model fails.

Despite the efficiency of our CCSD(T)*/CBS extrapolation scheme, optimal curation of the coupled-cluster data set is still essential since we can only perform a limited number of these calculations. As a source of structures for CCSD(T)*/CBS data generation, we choose to systematically subsample the existing data set with 5M molecules, since this data set already provides a pool of highly diverse molecular configurations and conformations. We begin with an initial random subsample of 200k data points, then iteratively we select new data for coupled-cluster calculations according to maximal ensemble disagreement (i.e., query by committee[64]). Through three iterations of coupled-cluster data generation using active learning, we grow the coupled-cluster data set to about 480k molecules. To further improve the ANI potential's description of torsion profiles, we also perform 20 iterations of active learning[25] on random molecular torsions from small and drug-like molecules to enhance ANI-1x with about 200k new DFT calculations. The ANI driven torsion sampling technique is detailed in Section S1.3. Of these torsion conformations, we randomly select 10% for CCSD(T)*/CBS calculations. The result is an enhanced ANI-1x DFT data set containing 5.2M data points and a high-accuracy CCSD(T)*/CBS data set containing about 500k data points.

**Training to high-accuracy data using transfer learning.** Here we describe the transfer learning methodology (depicted schematically in Fig. 4) used to create ANI-1ccx. First, an ANI potential is trained to the DFT data set with the new active learning torsion data added, yielding a potential equivalent to the ANI-1x potential[25]. Note, a single ANI potential is composed of multiple ANI neural network models. We then retrain each ANI-1x model to the CCSD(T)*/CBS data with 65,280 of the 325,248 optimizable neural network parameters held constant for each ANI model in the ensemble. Training a single ANI model to the original 5.2 million molecule data set takes ~4 h on a NVIDIA Titan V GPU, while retraining to the 500k molecule CCSD(T)*/CBS data set takes around 30 min. Neural network parameters are organized into a set of hidden layers. The ANI models trained in this work contain four hidden layers; we leave two hidden layers to be optimized

---

### Table 3 Computational cost and accuracy of our coupled-cluster approximation

| | CPU-core hours[a] | | Mean absolute deviation from CCSD(T)-F12 (kcal mol$^{-1}$) | |
|---|---|---|---|---|
| | **Alanine (13 atoms)** | **Aspirin (21 atoms)** | **S66** | **W4-11** |
| CCSD(T)/CBS | 9.13 | 427.00 | 0.03 | 1.31 |
| CCSD(T)*/CBS (this work) | 1.44 | 7.44 | 0.09 | 1.46 |

[a]All calculations are performed on an Intel Xeon E5-2630 v3 @ 2.40 GHz CPU

---

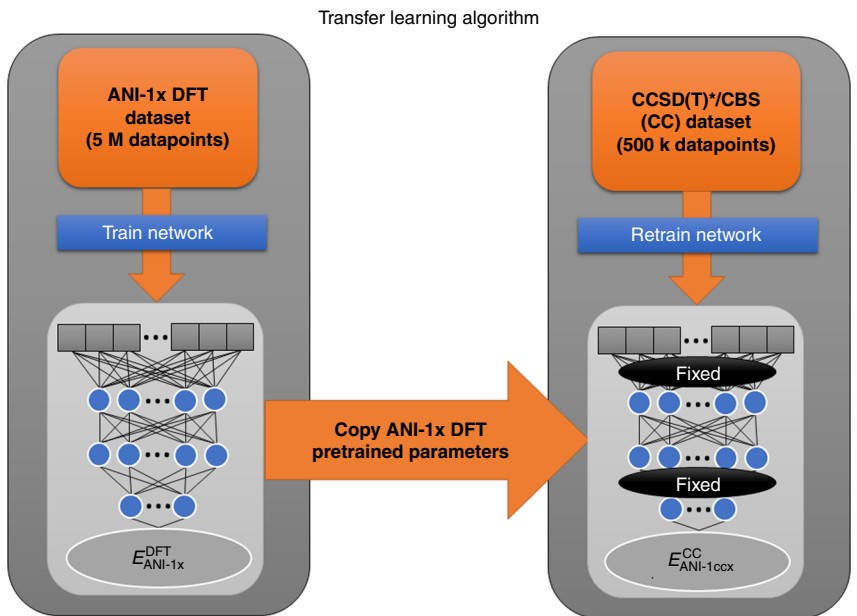

**Fig. 4** Diagram of the transfer learning technique evaluated in this work. Transfer learning starts from a pretrained ANI-1x DFT model, then retrains to higher accuracy CCSD(T)*/CBS data with some parameters fixed during training

during the transfer learning process, while the other two layers are left fixed to reduce the number of optimizable parameters during the training process and thus avoid overfitting to the smaller CCSD(T)*/CBS data set. Details of ANI-1ccx's performance on its test set are given in Supplementary Table 3. An alternative to transfer learning is Δ-learning[46]. With Δ-learning, one trains a new model to correct for the difference between CCSD(T)/CBS and the existing model pretrained on DFT data. Although Δ-learning yields similar accuracy to transfer learning, it needs to evaluate the neural networks twice to make inferences. More information on Δ-learning and its accuracy is provided in Supplementary Fig. 2 and Supplementary Table 4.

## Data availability

All relevant data are available from the authors upon reasonable requests.

## Code availability

All code needed to run this model can be found at https://github.com/isayev/ASE_ANI

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

## Acknowledgements

J.S.S. thanks the University of Florida for the graduate student fellowship and the Los Alamos National Laboratory (LANL) Center for Non-linear Studies for resources and hospitality. R.Z. acknowledges support by National Science Foundation (NSF) grants 1456638 and 1338192. We gratefully acknowledge the support and hardware donation from NVIDIA Corporation and express our special gratitude to Mark Berger. The authors acknowledge support of the U.S. Department of Energy (DOE) through the LANL LDRD Program. This work was performed, in part, at the Center for Integrated Nanotechnologies, an Office of Science User Facility operated for the U.S. DOE Office of Science. We also acknowledge the LANL Institutional Computing (IC) program and ACL data team for providing computational resources. O.I. acknowledges support from DOD-ONR (N00014-16-1-2311) and Eshelman Institute for Innovation award. The authors acknowledge Extreme Science and Engineering Discovery Environment (XSEDE) award DMR110088, which is supported by National Science Foundation grant number ACI-1053575. This research in part was done using resources provided by the Open Science Grid[65,66] which is supported by the National Science Foundation award 1148698, and the U.S. DOE Office of Science. A.E.R. thanks NSF CHE-1802831 and O.I. thanks NSF CHE-1802789.

## Author contributions

J.S.S. modified codes for transfer learning, assisted in data selection, analyzed results, formulated methods, and wrote the paper. B.T.N. developed scripts to generate CCSD(T)*/CBS data, ran all calculations, and edited the manuscript. R.Z. developed the CCSD(T)*/CBS extrapolation scheme, assisted in the analysis of results, and wrote the manuscript. N.L. helped formulate transfer learning methodology and edited the paper. C.D. wrote scripts for relaxed torsion scans using ANI models, assisted in benchmarks, and edited the manuscript. K.B. helped formulate methods and edited the paper. S.T., O.I., and A.E.R formulated methods, analyzed results, and edited the paper.

## Additional information

**Competing interests:** The authors declare no competing interests.

