## [Peer Review File · Nature Communications]

Reviewers' comments:

Reviewer #1 (Remarks to the Author):

The authors present a new interatomic potential (ANI-1ccx), based on the machine learning framework called ANI, which is previous work of some of the authors. The framework consists of a neural networks which is trained to predict energies of organic molecules.

The main difference between ANI-1ccx and its previous iteration, ANI-1x is that some layers of the neural network have been retrained to model CCSD(T) ("gold standard" accuracy of quantum chemical calculations, but very costly in computational effort) energies rather than DFT (affordable, but less accurate) energies, which is referred as "transfer learning" and exploits the assumption that the neural networks representing DFT and CCSD(T) potential energy surfaces are similar, but there is a lot less data available for the more expensive CCSD(T) level of calculations.

The authors provide extensive benchmarks of the new potential energy surface, including molecular energies, barriers etc.

Regarding the question whether this manuscript should be published in Nature Communications, I refer to the guidelines recommended for reviewers. I assume the result of the paper is the current parameterisation of ANI-1ccx, as ANI itself has been already published, and transfer learning, which is the new component of the training procedure seems to be a standard technique in neural network training.

Criteria for publication

To be published in Nature Communications a paper should meet several general criteria:

- The data is technically sound:

Yes, the authors used reputable software (ORCA) to generate the training and benchmark data

- The paper provides strong evidence for its conclusions

Yes, The authors provide extensive benchmarks of ANI-1ccx

The results are novel (we do not consider abstracts and internet preprints to compromise novelty)

- Sort of, this is a modified parameterisation of a previous model. There are also other similar models in the recent literature (cited in the manuscript) which also target CCSD(T) accuracy at a similar computational cost, although this current work is more extensive with its focus on reaction barriers and forces.

- The manuscript is important to scientists in the specific field

Yes, such an interatomic potential, if generally available, can enable calculations which, if using CCSD(T) directly, would be prohibitive computationally.

- In general, to be acceptable, a paper should represent an advance in understanding likely to influence thinking in the field.

I'm not entirely sure the paper represent an advance in understanding - it provides a potentially useful tool, a significant upgrade from the previous version of the work, which can be used in the future to produce papers like that, but this is a bit convoluted argument. I think it should be up to the editors to decide whether Nature Communications is the right journal to publish this work.

Reviewer #2 (Remarks to the Author):

Please see the attached document.

The following points highlight the changes to our manuscript titled: “Outsmarting quantum chemistry through transfer learning.” These changes are based on the feedback we received from the referees of our first submitted manuscript.

- As requested by the reviewers we add a discussion of limitations. Here we provide insights into opportunities for future research based on these limitations. (page 16)
- We have clarified our timing comparisons so that it will be clear to the reader that it is a qualitative comparison. Such a qualitative comparison is made difficult by the lack of freely available programs (i.e. OPLS3) and that DFT GPU code is just not efficient on small molecules. (page 15)
- We have clarified the model size (number of optimizable parameters) within the text. (page 8)
- We have clarified our use of an ensemble and the reasoning behind this usage. (page 9)
- We have added model training and retraining times as requested. (page 8)
- We have clarified the details of the torsional benchmark used in the work. (page 14)
- We added a better description of the “Whiskers” in the box plot (Figure 4).
- We have clarified the methods of restrained optimization used in the work. (page 14)
- Due to reviewer interest in condensed phase simulation we have added text to clarify the models capabilities in this space (page 16). We also include a condensed phase RDF in the supplemental (Figure S5) info as evidence that such models will one day be very applicable for condensed phase simulation

We thank all 3 reviewers for their very useful comments. We respond in detail to all their queries and point to the changes we have made in the manuscript.

Reviewer #1:

Comment: “Regarding the question whether this manuscript should be published in *Nature Communications*, I refer to the guidelines recommended for reviewers. I assume the result of the paper is the current parameterisation of ANI-1ccx, as ANI itself has been already published, and transfer learning, which is the new component of the training procedure seems to be a standard technique in neural network training.”

Response: The reviewer is correct that a primary contribution of this work is the ANI-1ccx potential, but we would like to add that the combination of a general-purpose ML potential developed using active learning and transfer learning methods has yet to be explored in existing literature. Developing machine learning methods that can effectively combine multiple sources of data is an outstanding problem for machine learning modeling of general-purpose atomistic potentials. We view the present work as a large step in that direction.

Comment: - *The data is technically sound:
Yes, the authors used reputable software (ORCA) to generate the training and benchmark data*

Response: We thank the reviewer.

Comment: - *The paper provides strong evidence for its conclusions
Yes, The authors provide extensive benchmarks of ANI-1ccx*

Response: Indeed, an important component of this work is the large and extensive set of tests on which we benchmark ANI-1ccx, where we push the systems studied far from the set of molecules used for parameterization.

Comment: *The results are novel (we do not consider abstracts and internet preprints to compromise novelty)
- Sort of, this is a modified parameterisation of a previous model. There are also other similar models in the recent literature (cited in the manuscript) which also target CCSD(T) accuracy at a similar computational cost, although this current work is more extensive with its focus on reaction barriers and forces.*

Response: We agree that the present work distinguishes itself by its extensive applicability, which is made possible partly by the transfer learning methodology. Further, the extensive CCSD(T)/CBS dataset featuring non-equilibrium molecular conformations generated through active learning will provide a very unique contribution to the community.

Comment: - *The manuscript is important to scientists in the specific field
Yes, such an interatomic potential, if generally available, can enable calculations which, if using CCSD(T) directly, would be prohibitive computationally.*

Response: We thank the reviewer and point out that we are making the model and the predictor programs available to the community upon publication of this work.

Comment: - *In general, to be acceptable, a paper should represent an advance in understanding likely to influence thinking in the field.
I'm not entirely sure the paper represents an advance in understanding - it provides a potentially useful tool, a significant upgrade from the previous version of the work, which can be used in the future to produce*

papers like that, but this is a bit convoluted argument. I think it should be up to the editors to decide whether Nature Communications is the right journal to publish this work.

Response: We thank the reviewer for sharing his/her perspective. In our view, the present manuscript offers myriad new contributions over typical literature in the field of machine learning for force fields (potential energies) development. *A key point of this paper is that ANI-1ccx has excellent transferability across a variety of realistic applications, which are well outside the training data. Most ML potentials are created to work only on one system at a time, while ours can, in principle, be used for any organic molecule without retraining.* Further, we explore a transfer learning-based methodology to achieve coupled cluster-level accuracy; we anticipate that the methods we introduce will be highly influential on future ML studies for molecular systems. For example, an outstanding challenge is training on both simulation and experimental data, for which transfer learning could be a promising methodology.

Reviewer #2

Comment: The authors present a new neural-network potential (ANI-1ccxx) and training methodology that retains most of the accuracy of high-level ab initio methods as a fraction of the computational cost. In departure from the earlier DFT-trained ANI-1x neural-network potential developed by the authors, a new method to train the new ANI-1ccxx model to an approximate coupled clusters method (CCSD(T)* / CBS) is created. The proposed method involved (1) generating an actively learned, DFT-based ANI-1x-style training set, (2) training an ensemble of ANI-1x-style potential to the aforementioned training set, (3) down-selecting configurations in that training set based on maximal ensemble disagreement, (4) re-computing the down-selected training set at the CCSD(T)* / CBS level, and (5) refitting half of the ANI-1x-style potential's hidden layers to form the new ANI-1ccxx model. The resulting ANI-1ccxx model is shown capable of serving as a high-quality proxy for coupled-cluster predictions of molecular conformers and gas-phase reaction energies. Overall, this paper presents a novel, and promising methodology for circumventing prohibitively expensive QM calculations on large molecules and represents a leap in the direction a machine-learned potential that can address the capability gap between first principles and molecular mechanics methods.

Response: We thank the reviewer for his/her supportive comments.

Comment: *The juxtaposition of statements like "A remaining challenge is to develop ML-based potentials that reach coupled cluster-level accuracy while retaining transferability and extensibility over a broad chemical space." and "The present work uses transfer learning to train an ML potential that is accurate, transferable, extensible, and therefore, broadly applicable," coupled with the authors definition of extensible and comment molecular mechanics potentials' failings being caused by the desire to get to larger scales suggests the authors believe ANI-1ccx can do so.*

It is perhaps a matter of semantics, but there seems to be confusion between large molecules and large systems (i.e. condensed phase simulations). ANI-1ccx performs well for large molecules in vacuum (though I assume self-solvating molecules have not been investigated), however there is no reason to believe ANI-1ccx would work well for any kind of solvated, condensed phase system. This needs to be clear to the reader.

Response: We thank the reviewer for raising this important point. We have added a section to the conclusion that makes this more clear:

"Future work will aim to validate and retrain (if necessary) the ANI-1ccx potential for applications in condensed phase simulation. For smaller molecular systems, the ANI-1ccx potential is an accurate and efficient alternative to expensive QM methods, and might find indirect ways to become applicable in such condensed phase simulation, e.g. using ANI-1ccx to parametrize force fields for condensed phase simulation."

We also want to clarify that the language of solvent/solute/condensed phases, while prevalent in the force field community (where molecules have bonding patterns and are separate units), is not that useful when using quantum mechanics. In QM, we only really have atoms in space, and all discussions about solvent/solute, bonding, charges are done after the calculation is done, by reading the electron density. Hence, in principle, our method can be used for condensed phases and solvation also. The question then becomes if, as-is, the potential can properly describe the interactions. That is an issue of data and training, but in principle, there is no formal barrier to do it.

As an example, we have added a radial distribution function (RDF) from a condensed phase simulation of liquid cyclohexane compared to experiment to the supplemental information (Figure S5). While ANI-1ccx might not be expected to work well here, it actually provides a qualitative comparison to the experimental RDF. We show this as an example that such simulations are technically feasible, and the results already show what part of the phase space one would need to explore and retrain for a better treatment. However, such an endeavor is outside of the scope of the present work.

Comment: Along the same thread, I suggest the authors be clear about what ANI-1ccx cannot do. At several points in the manuscript, there is discussion of the limitations of QM, MM, and even other ML potentials, but not of ANI-1ccx. For example, will ANI-1ccx natively work for condensed phase systems? Get excited state chemistry?

Response: We have revised the conclusions of the manuscript to comment on these limitations of ANI-1ccx:

“As with any model, ani-1ccx has limitations. Some of them can be overcome by adding more data and through active learning methods and retraining. This category includes new and different chemical environments, intermolecular interactions, ions, new atomic elements and reactions. There is a set of limitations that would require the development of new theory and methods, for instance for recovering long range interactions through the addition of coulomb interactions, to treat multiple electronic excited states or radicals.”

Comment: Discussion of computational efficiency is opaque. Benchmarks given on different machines and different compute architectures are meaningless. Any comparisons between QM, MM, and ANI need to be given on (1) on the same CPUs or the same GPUs, not some mix of the two (2) in terms of core-hours or core-minutes. It is one thing for a calculation to run faster than another, but if it takes 100x the computing power, that needs to be clear to the reader. Additionally, the text should mention how many parameters are contained in the ANI-1ccx model.

Response: We have revised the manuscript to clarify and sharpen our statements about the computational efficiency and number of parameters of ANI-1ccx. Universal comparisons are challenging to make, because 1) we do not have a license for OPLS3, so we opt to use published timings in the comparison, 2) No highly efficient and free to use DFT code for small molecules exists on GPUs, and 3) the implementation of our ANI model currently exists only on GPUs, as GPUs are the most efficient platform for such models. To give the reader a loose sense of ANI-1ccx performance, we compare ANI-1ccx running on a single NVIDIA V100 GPU to traditional classical force fields and *ab initio* calculations running on a single modern CPU. In this limited context, we conclude that ANI-1ccx is approximately 50 to 100x slower than classical force fields, and scales far better with molecular size than QM methods. We believe that such a comparison, though qualitative, can still be useful to a user of the ANI-1ccx potential and shows the practicality of its application over QM methods. This is reflected in the following statement toward the end of the results section:

“While a GPU to CPU comparison with the use of different optimization methods is not exactly a fair comparison, it does provide a sense of the computational affordability of the ANI potential.”

The number of model parameters is now given in the section describing the transfer learning technique on page 8.

“We then retrain each ANI-1x model to the CCSD(T)/CBS data with 65,280 of the 325,248 optimizable neural network parameters held constant for each ANI model in the ensemble.”*

Further information about the exact network architecture can be found in the supplemental information (see section S1.2.1).

We also copy the table describing the architecture here:

Hydrogen Network Architecture				
	Layer1	Layer2	Layer3	Layer4
Nodes	160	128	96	1
Activation	CELU ¹²	CELU	CELU	Linear
Regularization	L2 (1.0E-4)	L2 (1.0E-5)	L2 (1.0E-6)	None
Carbon Network Architecture				
	Layer1	Layer2	Layer3	Layer4
Nodes	144	112	96	1
Activation	CELU	CELU	CELU	Linear
Regularization	L2 (1.0E-4)	L2 (1.0E-5)	L2 (1.0E-6)	None
Oxygen and Nitrogen Network Architecture				
	Layer1	Layer2	Layer3	Layer4
Nodes	128	112	96	1
Activation	CELU	CELU	CELU	Linear
Regularization	L2 (1.0E-4)	L2 (1.0E-5)	L2 (1.0E-6)	None

Comment: The authors discuss using ensembles of models to decide which configurations to include in the final down-selected repository. Is ANI-1ccx itself an amalgam of machine learning potentials as well? Additionally, one might imagine that as ANI-1ccx grows in use, occasions may arise wherein ANI-1ccx needs to be retrained. How much effort would this take? Worded otherwise, how long does it take to retrain one of these potentials to an updated dataset?

Response: Yes, the ANI-1ccx potential itself is an ensemble of models. This is mentioned in two locations within the article. Specifically, the active learning section:

“To reduce variance and increase accuracy, all ANI results presented in this work are the ensemble prediction of 8 ANI neural networks, i.e. the ANI-1x potential used in this work is an ensemble of 8 ANI-1x neural networks trained to different splits of the ANI-1x dataset and the ANI-1ccx network is built by transfer learning from the eight ANI-1x networks.”⁴⁶

And within the results section:

“Recall that each ANI model is an ensemble average over 8 neural networks. Without an ensemble of networks, the MAD and RMSD of ANI models degrades by about 25%.⁶⁶”

As for retraining the ANI models, within the NeuroChem implementation used in this work retraining has little cost. The most costly part of retraining is the generation of new QM data. We have added the following text to the article with approximate timings in the “Training to high-accuracy data using transfer learning” section on page 8.

“Training a single ANI model to the original 5.2 million molecule dataset takes approximately 4 hours on a NVIDIA Titan V GPU, while retraining to the 500k molecule CCSD(T)/CBS dataset takes around 30 minutes.”*

Specific points:

Comment: Throughout the manuscript, “empirical,” and “classical” are used interchangeably to refer to force fields (FFs). An “empirical FF” simply means “trained to experimental data,” whereas “classical FF,” means follows classical mechanics. It is more likely that the authors mean the latter. As an example, The OPLS models the authors compare to are semi-empirical non-reactive classical force field.

Response: We have revised our manuscript for clarity. Our goal with the use of ‘empirical’ was to point out something non-QM and fitted, as many non-classical things from ML can fit in this category. Though, we agree that this use can be confusing. Instead, we now explicitly state “non-QM model potential” in various places to make this clearer and have referred to ‘classical force fields’ directly when needed.

Comment: How do the authors select 8 as an optimal number of networks in your “committee”?

Response: We selected 8 members based on an empirical trade-off between computational efficiency, reduction of variance, and efficacy of the ensemble uncertainty. Our revised manuscript (page 6) includes an additional citation to our previous work, where these considerations are discussed in more detail.

“To reduce variance and increase accuracy, all ANI results presented in this work are the ensemble prediction of 8 ANI neural networks, i.e. the ANI-1x potential used in this work is an ensemble of 8 ANI-1x neural networks trained to different splits of the ANI-1x dataset.⁴⁶”

Comment: On Page 10 the authors state “Accurate forces are important for MD simulations and geometry optimization. Therefore, we explicitly assess force accuracy as well.” What sort of MD simulations do the authors envision with the current ANI-1ccx model?

Response: Such forces could be used for geometry optimizations, MD simulations to generate IR spectra (DOI: 10.1021/acs.jpcclett.8b01939), or MD for small drug molecules. In terms of MD, we can use now for conformational sampling of small molecules, but once we move (a future publication) into condensed phases, we would like the forces to be as accurate as possible.

Comment: In figure 3, there are 4 instances where ANI-1ccx yields energy differences with the opposite sign predicted by CCSD. Are these differences meaningful? If so, can you speculate on the cause based on your training set and these specific validation cases?

Response: We thank the reviewer for their observation. The most likely answer is that the model is experiencing random error compared to the reference, and when the energy differences are small that random error can flip the sign. Another potential argument that is more difficult to prove is that the model is biased in a small way toward the original DFT training set, albeit greatly improved over DFT. For example, in both reactions 1 and 2 of panel 4 in Figure 3 ANI predicts an energy with an error biased towards DFT. However, this is speculative, and we opted to leave interpretation to the reader.

Comment: On page 13 the authors state “ANI-1ccx transfer learning-based potential against various QM and molecular mechanics (MM) based methods from the molecular (torsion benchmark of Sellers et. al.” Please provide the reader with some feel for what molecules make it in to this set.

Response: We altered the following sentence to make this clearer.

“This benchmark provides a measure of accuracy for a model at reproducing potential energy profiles from a diverse set of molecular torsions of small organic molecules containing the atoms C, H, N and O. These torsions are representative torsions typically found in small drug like molecules.”

Comment: *In figure 4, several data sets have maximum MAD values well outside the plotted “whiskers.” What do these data points correspond to? Additionally, what metric is used to draw these “whiskers”? Do they provide the standard deviation?*

Response: As stated in the comment of Figure 4: “... the “whiskers” guide the eye to identify outliers as in Sellers et al.” We have modified the text to include the following sentence:

“The upper “whisker” extends to the last datum less than the third quartile plus 1.5 times the interquartile range while the lower “whisker” extends to the first datum greater than the first quartile minus 1.5 times the interquartile range.”

To see more information on standard box plots see the following link. (NIST/SEMATECH. "Box Plot." §1.3.3.7 in NIST/ SEMATECH e-Handbook of Statistical Methods. <http://www.itl.nist.gov/div898/handbook/eda/section3/boxplot.htm>)

Comment: *Discussion of how the torsion benchmark is confusing: “The red QM methods (first three from the left) are computationally intense QM methods which used MP2 restrained optimized structures, while the green QM methods were all optimized using their own forces. The ANI and MM models also carried out restrained optimizations using their own forces. The ANI-Ix potential, trained to” First, what exactly is meant by “restrained optimized”? Additionally, it seems inconsistent to use structures computed at MP2 for certain methods, while other methods compute geometry based on their own gradients. Why did the authors not MP2 configurations for all methods? Do the authors expect that doing so would lead to a significant change in results?*

Response: We have modified the section quoted to clarify. See the text below:

“Each torsion in the benchmark is generated through a restrained optimization, where the torsional degree of freedom is fixed every 10 degrees, and the remaining degrees of freedom are relaxed through an optimization process. The red boxes in Figure 4 represent QM methods (first three from the left) that are so computationally intense the MP2 restrained optimized structures were used and single point calculations were performed for that method. The green QM methods were all optimized using their own forces. In an ideal setting, all methods would provide their own structures for the final energy calculation, though, such an exercise would be prohibitively computationally expensive for the more rigorous QM methods. The ANI and MM models also carried out restrained optimizations using their own forces. The MP2 structures are not used here because the usefulness of such efficient methods can only be gauged without the assistance of less efficient QM methods.”

Comment: *Wording of the concluding sentence “We conclude that subtle but important physics captured by gold-standard QM is modeled by ANI-Iccx” is confusing*

Response: The manuscript now states: *“After extensive benchmarking, we conclude that ANI-Iccx captures a broad range of organic chemistry, with accuracy comparable to QM calculations at the coupled-cluster level of theory.”*

Reviewer #3

Comment: *None of the above ideas are particularly new, although they have not been assembled in this particular way at this scale. Large scale fitting of molecular energies has been rather fashionable in the*

last couple of years, and in this sense "technology demonstrators", using larger datasets, higher levels of quantum chemistry, etc feel like an incremental advance.

Response: We thank the referee for this critique. Compared to previous work in this area, the other 2 Referees have recognized the novelty of the present study in 4 aspects:

1. An extended high fidelity coupled cluster dataset
2. Emphasis to training to non-equilibrium and torsional barriers
3. Broad extensibility and generality to other organic class systems
4. Demonstrated advantages of transfer learning as promising direction for other ML algorithms in chemistry and materials science
5. ANI-1ccx is free to use and provided to the community

In the revised manuscript we emphasized these points to improve the message and future impact of the article. In particular, the last paragraph of the introduction now covers all points made above. Also, please also see our response to the last comment of first Referee. Hopefully our editions clarified the value of our work for the reviewer and for the readers.

Comment: *The authors call their model a "force field". It is undeniable that empirical parametrisations of molecular energies and interactions have been the workhorse of soft matter modelling for decades, their development in terms of functional forms has reached a plateau, and new ideas to improve their accuracy are very important.*

Empirical force fields are often parametrised to explicitly reproduce important complex observables, such as equations of state, solvation enthalpies, etc. Their power comes from the fact that these properties appear to be the controlling quantities in predictions of scientific interest, such as properties of condensed phases, protein-drug interactions, etc. In contrast, the idea of fitting a model to a large quantum chemistry database is that if the potential energy surface is predicted accurately enough, all complex properties should follow. This has been the overarching principle of first principles modelling. Therefore the testing of such a model has to be in two steps. First, one could loosely call it "verification", it needs to be shown that the potential energy surface has indeed been fitted accurately, or in the present case, significantly more accurately than before. This is what the present manuscript does. Given the assembled proven techniques, this is not particularly surprising. In the second step, loosely called "validation" one needs to show that the complex properties of interest in force-field type simulations are predicted better than current force fields. The manuscript does not do anything in this regard.

Response: We would like to clarify what we mean by a force field, which is different than the reviewer's view. We think of this method as a force field simply in that it is a map from structure to energies. In our case, it does not have an explicit functional form. If now one thinks about this as a mapping function, then we aim to reproduce what quantum mechanics would give for the same geometry, without trying to reproduce or predict condensed phase properties. This is very different than traditional force fields, where one would adjust the parameters not just to the energy, but to reproduce actual macroscopic observables. Benchmarking against the previous state-of-the-art methods is an extremely important component of this work. We compared ANI-1ccx to many other methods, including CCSD(T), a gold standard for quantum chemistry in various real-world applications. The comparisons employed are established test cases and methods that have been trusted for DFT model validation for many years. To further ensure that the potential is at least qualitative in bulk simulation we have included results in the supplemental information (Figure S5) from a simulation on cyclohexane (NVT @ 300K for 50ps) in bulk phase including a comparison with experimental results (please see also our response to Reviewer 1). Even though this potential was not trained for such a simulation, it is well behaved enough to qualitatively reproduce the RDF. This represents a test case that far exceeds anything the model was trained to do. Further validation against condensed phase experimental values would warrant a completely new publication as the methods, new data generation, and results would reach that level. We want to emphasize that to the two other reviewers, the applications shown in the current article are sufficiently impactful to warrant publication in Nature Communications.

Comment: *What comes the closest are the torsional maps - these are still direct energies (differences), but their key role in conformational flexibility makes them important, and their accurate description is not directly implied by reproduction of absolute energies to the same accuracy. It is notable that this is where "iterative fitting" was used, underscoring the point. Therefore, the convergence of this procedure is important. Do the authors claim that their final fit now reproduces torsional barriers for all of their molecules with a similar accuracy? The 45 tests in Fig 4 certainly show an impressive improvement over OPLS, and it is implied that these tests are different from those that were put into the fitting during the iterative fitting.*

Response: Iterative fitting shows that even in cases where the potential might not perform as well as desired, it can be improved in a fully automated fashion. We include results from the potential prior to the iterative fitting in the supplemental information. These results show that even our prior model outperforms the industry standard force field (OPLS3), without explicitly fitting to torsions. Considering that all force fields designed to work on small druglike molecules (e.g. OPLS3) are likely fit to exactly these torsions as well (since the molecules represent the smallest possible unit of such a torsion), this comparison does not seem so unfair. Our previous work (ANI-1/ANI-1x) shows that when a model is trained well for small molecule interactions, those interactions are improved in larger systems. Ideally, we would run coupled cluster calculations on torsions of 50+ atom structures to demonstrate extensibility, but this is out of scope for the present work.

Comment: *The fitting of models in a large dimensional space, such as the present model, is notorious for giving smooth and accurate results for configurations near the fitting set and becoming much less accurate and rough away from it. Therefore, instead of reporting just torsional energies as averages, it is very important to show torsional curves, and two-dimensional torsional maps (using two torsion angles), as is standard in the force field literature.*

Response: Our previous literature on the ANI style of building ML potentials has shown that, with the correct dataset, one can maintain a smooth and accurate potential across configurational and conformational space of organic molecules, especially within a specific temperature range of interest. The ANI-1ccx potential developed in this work enabled a restrained optimization along all 36 points of each of the 45 torsions, which speaks to the "smoothness" of the model. Were the potential not smooth, optimization would produce a bad structure with a very poor energy prediction. The low errors in Figure 4 and SI Figure S1 speak to the global accuracy and smoothness of ANI-1ccx.

Comment: *In the same spirit, the high dimensional nature of the model worries me quite a bit. The fragility of artificial neural network models (for example in image recognition) is now well known: they give essentially random results for inputs that are different from the those in the training dataset in small but important aspects. While in standard machine learning tasks, this does not necessarily prevent effective use, in molecular modelling it does: if a configuration that has high real energy (meaning it and similar configuration do not appear in the training set) is predicted to have a much lower energy, such configurations will be overrepresented when sampling is done with the model. Such "holes" in the potential energy landscape plagued early models of molecular potential energy surfaces even in moderate (10-20) dimensions. The authors need to do extensive sampling with their model (molecular dynamics, Monte Carlo at high temperatures, or simply geometry optimisation from randomised starting points) and show that the energy of the resulting structures are either predicted accurately, or at least not significantly underpredicted compared to the quantum mechanical target.*

Response: The problem of high-dimensionality leading to "holes" in a potential has indeed been a concern in the ML potential community for many years. The purpose of using active learning in the development of our datasets aims to mitigate this problem. Further the combination multiple techniques allow us to further minimize such problems:

1. Active learning search for 'holes'
2. Ensemble predictions

3. Reduction of the neural network dimensionality for the transfer learning models
4. Regularization of the networks.

We accomplish point 1 with *active learning*, in which the training dataset and ML potential are constructed iteratively. An example of one such active learning sampling technique employed is where the ML potential is used to drive molecular dynamics simulation; new points are added to the training dataset whenever the ML model is uncertain about its predictions. With enough active learning, the ML potential can make uniformly good predictions across the entire space of sampled trajectories.

As evidence that models built using these techniques are capable of an accurate description of high dimension physics, in a previous work we ran 1ns NVT MD simulations at 300K on various large drug molecules and small proteins. This study produced a model with very high accuracy force predictions on systems with hundreds to thousands of degrees of freedom. See that work here:

(<https://aip.scitation.org/doi/abs/10.1063/1.5023802>)

Comment: *Lastly, the authors do not say anything about their treatment of long-range electrostatics. It would appear that their interactions have a finite cutoff of 5.2 Å. It is well known that this is a critical part of many empirical force fields, and the last significant development was the introduction of explicit polarisability. How do they hope to describe equations of state and other bulk properties of molecular liquids, e.g. water? A poor description of bulk water would seem to prevent the correct description of solvation enthalpies, without which force fields are not much use. But water is just the most prominent example, there are many others which critically depend on medium and long-range interactions.*

Response: Incorporation of long-range Coulomb interactions into ML potentials is an exciting topic, and the subject of ongoing research. In the present manuscript, we focus on organic chemistry applications that do not involve strong long-range interactions. We have added the following paragraph to the end of the conclusions:

“There is a set of limitations that would require the development of new theory and methods, for instance for recovering long range interactions through the addition of coulomb interactions, to treat multiple electronic excited states or radicals.”

Comment: *In summary, the authors have done a significant amount of work towards fitting an accurate model of molecular potential energies. This kind of work is necessary on the long road towards better force fields. I encourage them to do the kinds of tests that I described above to entice users of force fields that quantities of interest to them are better described. High profile publication of a "new force field" is then warranted.*

Response: We thank the reviewer for their comment. Force fields are used in a variety of applications which do not reach the level of direct comparisons to experimental condensed phase properties. For example, torsional scans and thermochemistry are regularly employed by computational drug designers in their search for novel drugs. The methods employed in this work might also have impact in future materials studies along the same line.

Comment: *I commend the authors for making their model available for testing. For the technical community of force field builders, it is equally important to publicly release the database to which the model was fitted. Only then can the procedure be checked and verified independently, the relationship of the database and the tests scrutinised, and the demonstrated techniques built upon.*

Response: We agree. We have begun a declassification process through Los Alamos National Laboratory. Once this process is complete, and the article is published, the CCSD(T)*/CBS data will be made available to the public.

Reviewers' comments:

Reviewer #2 (Remarks to the Author):

I am satisfied with the authors' responses to my comments. When taken holistically (i.e. my review in concert with those from the other two reviewers), the primary remaining concern appears to be whether the work presented is novel enough. Introduction of transfer learning to the ANI method may indeed be incremental, but it is worth noting that doing so has led to a -substantially- improved model. In that regard, generating a force field capable of coupled cluster accuracy for -arbitrary- molecular systems is certainly a novel achievement.

Reviewer #3 (Remarks to the Author):

The authors have done a lot to address the comments and criticisms of all reviewers. We still have a number of disagreements, but for the most part they are not substantial enough to hold up the paper. For example, it appears that I still failed to get across the idea that although ab-initio PES fits are indeed not fit to agree with experiments or even calculated macroscopic observables, that has to be the ultimate judgement of their value, and showing how accurately they reproduce the ab initio PES is not a substitute for that. Showing the macroscopic predictions in fact tells us to what accuracy the ab initio PES needs to be matched in order for the macroscopic prediction to work.

The one case where I can't let the authors off the hook is the regularity (smoothness) of the PES. The fact that geometry optimisation "works" is very weak evidence that the PES is regular, modern optimisers are very good. So the authors must include all 45 torsion curves in the supplementary, so that we can see how regular they are, to what extent they are free form artefacts which might arise from the high dimensionality of the fit.

Gabor Csanyi

Response to Reviewers:

Reviewer 2 is now satisfied with the new version, while reviewer 3 requests that “The authors must include all 45 torsion curves in the supplementary, so that we can see how regular they are, to what extent they are free form artefacts which might arise from the high dimensionality of the fit.”

The supplemental information now includes all 45 torsional scans done with our ani-ccx potential. We believe the reviewer will agree that the scans are indeed smooth. We also submit a file with the actual data used to plot those scans, in case readers want to create the plots themselves.

REVIEWERS' COMMENTS:

Reviewer#3

Thank you for putting the torsional curves into the supplementary. Most of them look nice and smooth, with just a few showing some small kinks - it would have been nice to include the benchmark CCSD(T)/CBS curves, which you have since you report the MAE with respect to them. I encourage you to do that in the proof stage.

I can recommend the paper for publication.